# *NADP-Dependent Malic Enzyme* Genes in Sweet Pepper Fruits: Involvement in Ripening and Modulation by Nitric Oxide (NO)

**DOI:** 10.3390/plants12122353

**Published:** 2023-06-17

**Authors:** Jorge Taboada, Salvador González-Gordo, María A. Muñoz-Vargas, José M. Palma, Francisco J. Corpas

**Affiliations:** Department of Stress, Development and Signaling in Plants, Group of Antioxidants, Free Radicals and Nitric Oxide in Biotechnology, Food and Agriculture, Estación Experimental del Zaidín (Spanish National Research Council, CSIC), C/Profesor Albareda, 1, 18008 Granada, Spain; jtaboada@correo.ugr.es (J.T.); salvador.gonzalez@eez.csic.es (S.G.-G.); mangeles.munoz@eez.csic.es (M.A.M.-V.)

**Keywords:** cis-regulatory element, fruit ripening, malate, NADPH, NADP dehydrogenases, nitric oxide, pepper

## Abstract

NADPH is an indispensable cofactor in a wide range of physiological processes that is generated by a family of NADPH dehydrogenases, of which the NADP-dependent malic enzyme (NADP-ME) is a member. Pepper (*Capsicum annuum* L.) fruit is a horticultural product consumed worldwide that has great nutritional and economic relevance. Besides the phenotypical changes that pepper fruit undergoes during ripening, there are many associated modifications at transcriptomic, proteome, biochemical and metabolic levels. Nitric oxide (NO) is a recognized signal molecule with regulatory functions in diverse plant processes. To our knowledge, there is very scarce information about the number of genes encoding for NADP-ME in pepper plants and their expression during the ripening of sweet pepper fruit. Using a data mining approach to evaluate the pepper plant genome and fruit transcriptome (RNA-seq), five *NADP-ME* genes were identified, and four of them, namely *CaNADP*-*ME2* to *CaNADP*-*ME5*, were expressed in fruit. The time course expression analysis of these genes during different fruit ripening stages, including green immature (G), breaking point (BP) and red ripe (R), showed that they were differentially modulated. Thus, while *CaNADP*-*ME3* and *CaNADP*-*ME5* were upregulated, *CaNADP-ME2* and *CaNADP-ME4* were downregulated. Exogenous NO treatment of fruit triggered the downregulation of *CaNADP-ME4*. We obtained a 50–75% ammonium–sulfate-enriched protein fraction containing CaNADP-ME enzyme activity, and this was assayed via non-denaturing polyacrylamide gel electrophoresis (PAGE). The results allow us to identify four isozymes designated from CaNADP-ME I to CaNADP-ME IV. Taken together, the data provide new pieces of information on the CaNADP-ME system with the identification of five *CaNADP-ME* genes and how the four genes expressed in pepper fruits are modulated during ripening and exogenous NO gas treatment.

## 1. Introduction

NADPH (reduced nicotinamide adenine dinucleotide phosphate) is a central cofactor in a wide range of biosynthetic pathways that are essential in the metabolism of plant cells, such as the Calvin cycle, carotenoids, fatty acids and the aromatic amino acids (Phe, Tyr, Trp) as well as proline [1,2,3]. Furthermore, NADPH also sustains cellular detoxification and defense; for example, it is necessary to support glutathione reductase activity as part of the ascorbate–glutathione cycle and the generation of superoxide radicals by NADPH oxidase (NOX), which is also designated as the respiratory burst oxidase homolog (Rboh) [4,5,6]. However, NADPH is also necessary for nitric oxide (NO) generation through L-arginine-dependent NO synthase-like activity [7]. 

The NADP-dependent malic enzyme (NADP-ME, EC 1.1.1.40) is one of the NADPH-generating dehydrogenases that catalyzes the reversible oxidative decarboxylation of L-malate using NADP^+^ as a coenzyme to generate pyruvate, CO_2_ and NADPH in the presence of a bivalent cation, either Mg^2+^ or Mn^2+^. NADP-ME plays multiple roles in plants [8,9], including in seed germination [10], development [11] and fruit physiology [12,13,14,15,16], as well as in the mechanism of response against diverse environmental stresses [17] including salinity [18,19,20], water stress [21,22], mechanical wounding [23], arsenic [24,25,26], chromium [27], herbicides [28], low temperatures [29] and potassium deficiency [30,31].

Pepper (*Capsicum annuum* L.) is an important horticultural crop whose fruit is consumed worldwide. It is characterized by a significant quantity of vitamin C, provitamin A and minerals such as calcium as well as other compounds with health-promoting properties [32]. Recently, it has been shown that during the ripening of pepper fruits, there is a noteworthy metabolism of reactive oxygen and nitrogen species (ROS and RNS, respectively), which has been associated with physiological nitro-oxidative stress, where different ROS-generating and antioxidant systems are differentially modulated [33,34,35,36,37,38,39,40]. Previous studies have described that NADP-ME activity is modulated during the ripening of pepper fruit [41,42], but, to our knowledge, there is not any information about the modulation of genes that code for NADP-MEs in pepper fruits. Therefore, the main goals of this study were (i) to identify the number of NADP-ME genes present in pepper plants; (ii) to determine which of these genes are expressed in the fruit and how they are modulated during ripening and after the exogenous application of NO; and (iii) to identify the number and the relative abundance of the NADP-ME isoenzymes.

## 2. Results

### 2.1. Identification of the NADP-ME Genes in Pepper (Capsicum annuum L.): Sequence and Cis-Regulatory Elements

Based on the available NADP-ME sequences in *Arabidopsis thaliana*, the analysis of the *C. annuum* L. genome has allowed us to identify five *NADP-ME* genes designated from *CaNADP-ME1* to *CaNADP-ME5* distributed in chromosomes (Chr) 3, 5, 8, 9 and 12, respectively. On the other hand, through data mining of the transcriptome obtained for sweet pepper fruits [33], four of these genes, *CaNADP-ME2* to *CaNADP-ME5*, were identified, as indicated in red color in Table 1. Some properties of these five genes and their corresponding encoded proteins, including the number of amino acids (aa), molecular mass (kDa) and their putative subcellular localization, are also recapitulated (Table 1).

The analysis of the genomic organization indicated that *CaNADP-ME1*, *CaNADP-ME3* and *CaNADP-ME5* comprise 20 exons and 19 introns, whereas *CaNADP-ME2* and *CaNADP-ME4* have 19 exons and 18 introns. What is remarkable is that the length of the introns is very different (see, for example, *CaNADP-ME1* and *CaNADP-ME4*), causing the length of the corresponding genes to oscillate between 5000 and 9000 nucleotides (Figure 1). 

As part of the characterization of these genes, the identification of cis-regulatory elements in 1500 regions upstream of the transcription starting point of the *CaNADP-ME* genes was accomplished. Figure 2 displays the heatmap analysis of 24 cis-regulatory elements of several families of elements involved in various processes, including (i) light response; (ii) stress; and (iii) phytohormones. Nevertheless, ABRE (ACGT-containing abscisic acid response element), which is involved in abscisic acid (ABA) responsiveness, was the cis-regulatory element that exerted the most remarkable positive effect on *CaNADP-ME4*. This was followed by Box4, as part of the light-responsive family, which positively affected *CaNADP*-*ME2* and *CaNADP-ME5*.

### 2.2. NADP-ME Proteins from Pepper: Sequence, Phylogenetic Analysis and Modeling

The protein analysis of the identified CaNADP-MEs (Table 1) indicated that the subunit molecular mass ranged from 64 kDa to 71 kDa, and they were distributed in the plastid and cytosol. The analysis of the primary structure of these CaNADP-MEs and their alignment allowed us to discriminate ten amino acid motifs. Figure 3a shows the distribution of these motifs in the different isozymes and Figure 3b illustrates the sequence of amino acids motifs where the height of each amino acid symbol is proportional to the degree of conservation in the consensus sequences. Based on previous information on the plant NADP-ME family [43,44,45], motif 1 contains the sequence VYTPTVGEAC, which corresponds to NADP binding site 1. Nevertheless, a second putative NADP binding site identified in maize NADP-ME corresponds to the sequence ILGLGDLGC, and it is also present in motif 2. On the other hand, motif 3 includes three residues involved in the metal binding: E229, D330 and D354 (using the numbering of NADP-ME1). Appendix A shows the amino acid alignment of these five CaNADP-MEs. The conserved residues involved in the binding of NADP and metal (Mg^2+^ or Mn^2+^) are indicated with boxes.

The phylogenetic analysis of the NADP-MEs from several plant species, including pepper (*Capsicum annuum*), *Arabidopsis thaliana*, maize (*Zea mays*), grape (*Vitis vinifera*), bean (*Phaseolus vulgaris*) and rice (*Oryza sativa* subs. *japonica*), among others, allowed the identification of four main groups, designated as I, II, III and IV, which are represented by different colors in Figure 4. The CaNADP-MEs found in pepper fruit are indicated in red. Group I, which includes CaNADP-ME2 and CaNADP-ME4, consists of both monocot and dicot plants. Groups III (CaNADP-ME1 and CaNADP-ME5) and IV (CaNADP-ME3) comprise dicot plants. On the other hand, group II is exclusively composed of monocots. 

NADP-ME is considered a tetrameric enzyme [46], and Figure 5 depicts the model of the quaternary structure of plastidial CaNADP-ME3 present in pepper fruits, where it marks the binding sites for NADP and metals in each one of the subunits, which are colored in yellow, green, cyan and dark blue.

### 2.3. Fruit CaNADP-ME Genes: Expression during Ripening and Exogenous NO Treatment

The analysis of the RNAseq of sweet pepper fruits of the four CaNADP-ME genes was carried out at different ripening stages and after their exposure to exogenous NO gas. Figure 6 illustrates the experimental design, which included green immature (G), breaking point (BP1) and red ripe (R) fruits. Additionally, for the exposure to exogenous NO gas, two additional groups were established: fruits treated with 5 ppm NO for 1 h (BP2 + NO) and another group that was not treated with NO (BP2 − NO), corresponding to the control group. 

Figure 7 shows the time course analysis of the four genes identified in pepper fruits. CaNADP-ME3 and CaNADP-ME5 were upregulated during fruit ripening, whereas CaNADP-ME2 and CaNADP-ME4 were downregulated. Exogenous NO treatment of fruit caused the downregulation of CaNADP-ME3 and CaNADP-ME4, but the other two genes were unaffected.

### 2.4. Identification of the CaNADP-ME Isozymes in Pepper Fruits

To obtain deeper insights into the characterization of the CaNADP-ME enzyme system in pepper fruits, an in-gel analysis of the activity was carried out. Due to previous assays showing the presence of very weak activity bands in crude extracts from pepper fruits, enriched 50–75% ammonium–sulfate protein fraction could be achieved. Figure 8 illustrates the enzymatic pattern of the CaNADP-ME isozymes detected in green pepper fruits. Using more than 240 μg of protein of this enriched fraction, four isozymes were detected in polyacrylamide gels and were designated as CaNADP-ME I to CaNADP-ME IV, according to their level of electrophoretic mobility. CaNADP-ME III and IV were the most abundant isozymes, representing 33 and 26% of all isozymes, respectively. They were followed by CaNADP-ME II (24%) and CaNADP-ME I (17%). Unfortunately, in ripe (red) fruits, despite trying multiple precipitation conditions, due to the nature of the plant materials, it was not possible to obtain a precipitate-enriched fraction with ammonium sulfate, so isozymatic analysis of NADP-ME was only accomplished for green fruits.

## 3. Discussion

Higher plants contain several NADP-ME isozymes with different subcellular locations, which play different functions. Usually, they are split into photosynthetic and non-photosynthetic isozymes according to their main physiological functions. Perhaps the most well-known function of chloroplast NADP-ME is its participation in the photosynthesis of the bundle sheath cells of C_4_ plants, which fix CO_2_ into a molecule with four carbon atoms before starting the photosynthetic Calvin–Benson cycle [9,47,48], and also in Crassulacean acid metabolism (CAM) plants [49]. On the other hand, the designated non-photosynthetic NADP-MEs are present in the plastid and cytosol of all types of plants (C_3_, C_4_ and CAM). Other functions of NADP-MEs are related to their involvement in malate equilibrium for stomatal movement [50] or pH regulation [51]. However, the generated NADPH is also needed for redox homeostasis [52,53] and in the mechanism of defense against diverse environmental stresses [1,29,54]. For example, in the response of Arabidopsis to the necrotrophic bacterium *Pectobacterium carotovorum*, it has been shown recently that NADP-ME2 is phosphorylated by the RPM1-INDUCED PROTEIN KINASE (RIPK) to increase the content of cytosolic NADPH, and this allows for the production of superoxide radicals by the respiratory burst oxidase homolog D (RbohD) [6].

### 3.1. The Pepper Genome Contains Five CaNADP-ME Genes but Only Four Genes (CaNADP-ME2 to ME5) Are Expressed in Fruits

In previous studies, the presence of NADP-ME in pepper fruits was detected, where the activity increased during the fruit ripening. This suggested that this activity might be associated with the supply of NADPH during this process [41,42], although it is also necessary for other enzymatic systems such as the Rboh, whose activity increases [34], as well as for lipid biosynthesis. However, to our knowledge, there is no information about the number of genes and isozymes of NADP-ME present in this non-climacteric fruit. The obtained data indicate that among the five *CaNADP-ME* genes detected in the pepper genome, only four were expressed in fruits that were differentially regulated during ripening and under an enriched NO atmosphere. 

The number of exons/introns found in pepper *CaNADP-ME* genes is quite similar to that described for other higher plants. In the case of cytosolic *NADP-ME* from beans (*Phaseolus vulgaris* L.), the DNA sequencing of this unique gene allowed the identification of 20 exons [55]. In *Arabidopsis thaliana*, which contains four *NADP-ME* genes, it was found that *AtNADP-ME1* and *AtNADP-ME2* have 19 exons, but *AtNADP-ME3* and *AtNADP-ME4* have only 18 exons [56]. In maize (*Zea mays*), the *NADP-MEs* showed 19 encoding exons, except *cNADP-ME2*, which had 8, because exons 12–19 were fused [45]. Intron sequences have multiple functions, such as regulating alternative splicing, enhancing gene expression, controlling mRNA transport or chromatin assembly, etc. [57,58]. However, in mammal genomes, it has been suggested that an alternation occurs between the oldest exons, that is, the most conserved and shorter introns. Conversely, longer introns tend to be between exons that are more contemporary [57]. In pepper, the observed difference in the intron length among the *CaNADP-ME* genes, which can be up to 4000 nucleotides, is remarkable. Considering this, it could be inferred that *CaNADP-ME1*, which contains the shortest introns, could have the oldest exons.

The encoding NADP-MEs share a high degree of identity with other plant NADP-MEs, and they have distinctive domains [45]. Furthermore, the four CaNADP-MEs identified in fruits were predicted to be located in the cytosol (CaNADP-ME2) and in the plastids (CaNADP-ME3 to CaNADP-ME3 5), which is analogous to other plant species [9,47,59,60]. Thus, the model plant Arabidopsis contains four *NADP-ME* genes in its genome, and three encode for cytosolic isozymes (NADP-ME1 to NADP-ME3), whereas NADP-ME2 is located in plastids [61].

The available information on the NADP-ME in edible fruits is scarce. In the climacteric tomato fruits, the occurrence of NADP-ME has been associated with the relevance of malate in starch metabolism, thus affecting the fruit aging and postharvest softening [12,16,62]. In grape berries (*Vitis vinifera* L.), the NADP-ME contributes to the accumulation of malate during ripening [63]. In zucchini fruits, storage at 15 °C for 2 days, followed by storage at 4 °C for 14 days, attenuated chilling injury in the fruits, and the analysis of *NADP-ME* gene expression showed an increase that was accompanied by a concomitant enhancement of *G6PDH* (*glucose-6-phosphate dehydrogenase*) gene expression as well as an increase in the battery of antioxidant enzymes [64].

Metabolic analyses have shown that organic acids including malate and citrate are accumulated in a broad range of climacteric and non-climacteric fruits [65]. Thus, in some climacteric fruits, malate is used as a substrate during the respiratory burst, whereas in non-climacteric fruits, malate is continually accumulated during ripening [66]. During hot pepper (*Capsicum chinense*) fruit development and ripening, malate content decreases during later developmental stages but it then increases during ripening [67]. Thus, a correlation between malate levels and genes involved in the synthesis of starch has been suggested [68]. A similar correlation was observed for genes associated with cell wall pathways and protein degradation [16]. Although these studies did not analyze the NADP-ME activity, it could be assumed that this activity was involved in the total malate pool considering that the reaction catalyzed by this enzyme is reversible (L-malate ↔ pyruvate). Furthermore, these analyses could be puzzling if the malate–oxaloacetate shuttles are also considered, as a mechanism that involves several dicarboxylate translocators and malate dehydrogenase isozymes that catalyze the reversible interconversion of malate and oxaloacetate using either NAD or NADP, which are located in chloroplasts, mitochondria and peroxisomes, thus allowing the modulation of the ATP/NAD(P)H ratio [69]. Furthermore, malate metabolism also connects chloroplasts and mitochondrial ROS production [70]. All these organelles undergo drastic metabolic changes during pepper fruit ripening [71,72,73].

### 3.2. During Fruit Ripening, the Expression of CaNADP-ME2 and ME4 Is Downregulated, Whereas That of CaNADP-ME3 Is Upregulated, and Exogenous NO Gas Exerts a Negative Effect on CaNADP-ME3 and ME4

To our knowledge, there is little available information on NO and NADP-ME at the gene level. However, in a previous study, we found through the use of in vitro assays that the NADP-ME activity of sweet pepper fruit was inhibited to various degrees in the presence of NO as well as hydrogen sulfide (H_2_S) donors [42]. This suggests that this enzyme could undergo post-translational modifications mediated by these two molecules, including nitration, *S*-nitrosation and persulfidation [74]. This is supported by proteomic studies on Arabidopsis where NADP-ME was identified to be a target of *S*-nitrosation [75] and persulfidation [76], although in any case, the effect on the enzyme activity has been investigated. 

More recently, using a recombinant protein of the Arabidopsis cytosolic NADP-ME2 protein (NP_196728.1), mass spectrometry analyses corroborated that Tyr73 could be nitrated and, consequently, provoke the disruption of the interactions between the specific amino acids responsible for protein structure stability [29]. This equivalent Tyr residue is present in all the CaNADP-MEs and corresponds to Tyr103 in CaNADP-ME3; however, in a previous study, preincubation of an enriched 50–75% (NH_4_)_2_SO_4_ protein fraction obtained from green pepper fruits showing NADP-ME activity with the nitrating peroxynitrite did not show any inhibitory effect [42], as it did in the Arabidopsis NADP-ME2. However, this apparent contradictory behavior could be explained by the fact that the enriched fruit fraction contained other NADP-MEs, and each was affected to a different degree. To corroborate this, it would be necessary to obtain the corresponding recombinant NADP-ME from pepper fruits and perform the corresponding assays, an issue which is being addressed in our laboratory.

## 4. Materials and Methods

### 4.1. Identification of the NADP-ME Genes in Pepper (Capsicum annuum L.), Chromosomal Location and Synteny Analysis

Different methodologies were used to ascertain the different NADP-ME-encoding genes in pepper. First, the pepper transcriptome and proteome were downloaded from the NCBI database at https://www.ncbi.nlm.nih.gov/sra/PRJNA668052 (accessed on 13 February 2023), and https://www.ncbi.nlm.nih.gov/bioproject/814299 (accessed on 13 February 2023), respectively). The amino acid sequences of NADP-MEs described in Arabidopsis thaliana were also downloaded from the UnirProtKB database. These sequences were used as a query to search for NADP-MEs in the complete pepper proteome using the BLASTP tool. Second, the InterProScan tool was used [77] to confirm the presence of the conserved regions of the malic enzyme, an N-terminal domain (IPR012301) and the NAD(P) binding site (IPR012302).

Location coordinates of the CaNADP-MEs identified in the pepper genome were obtained from the NCBI database. 

### 4.2. Phylogenetic and Conserved Motif Analyses of NADP-ME Protein Sequences

The identified NADP-ME protein sequences in pepper were used to construct a phylogenetic tree using the NADP-ME present in several plant species (see Appendix A). The alignment of NADP-MEs was performed using the CLUSTALW method [78]. Then, the aligned sequences were subjected to MEGA11 [79] to perform an unrooted maximum likelihood phylogenetic tree with default parameters. Finally, the resulting phylogenetic tree was modified using the online tool Evolview v3 [80]. Conserved motifs of CaNADP-MEs were analyzed using the MEME tool [81] and visualized using TBtools software v1.108 [82]. Protein localization was predicted based on their amino acid sequences and using several predictors: WoLF PSORT [83] and Plant-mSubP [84].

### 4.3. Introns/Exons and Cis-Regulatory Elements Analysis of the CaNADP-ME Genes

The distribution of introns and exons in the CaNADP-ME genes was obtained from the NCBI database. This information was obtained using the ‘Basic Biosequence View’ tool of the TBtools v1.108 software [82].

To predict putative promoter sequences of the identified CaNADP-ME genes obtained from the NCBI Nucleotide database (https://www.ncbi.nlm.nih.gov/nucleotide/; accessed on 10 February 2023), a 1500 bp region upstream of the transcription starting point of each gene was considered. These sequences were searched for possible cis-acting regulatory elements using the PantCARE tool [85]. These results were manually processed and visualized using the ‘Basic Biosequence View’ function of TBtools v1.108 software [82]. 

### 4.4. Plant Material and Exogenous Nitric Oxide (NO) Gas Treatment

Sweet pepper (*Capsicum annuum* L. cultivar Melchor, California-type) fruits were collected from plants grown in commercial plastic-covered greenhouses (El Ejido, Almería, Spain). Fruits without any external apparent injury were selected at three developmental stages: green immature (G), breaking point (BP1) and red ripe (R). The harvested fruits were placed in black plastic bags, transported to the laboratory at room temperature, washed with distilled water and kept at a low temperature (about 7 °C ± 1 °C) for 24 h. To evaluate the effect of the exogenous NO gas treatment, we set up two additional groups: fruits treated with 5 ppm NO for 1 h (BP2 + NO) and another group that was not treated with NO (BP2 − NO). After 3 days at room temperature, all fruits were chopped into small cubes (5 mm/edge), frozen under liquid nitrogen and stored at −80 °C until use. Figure 6 shows the experimental design followed in this study with the representative phenotypes of sweet pepper fruits at the different ripening stages and subjected to NO treatment [33].

### 4.5. Library Preparation and RNA Sequencing

All procedures were performed as previously described in [33] with minor modifications. Briefly, libraries were prepared using an Illumina protocol and were sequenced on an Illumina NextSeq550 platform using 2 × 75 bp paired-end reads. These reads were preprocessed to remove low-quality sequences. Useful reads were mapped against the set of transcripts available for C.annuum species in the NCBI database (assembly UCD10Xv1.1; accessed on 10 February 2023) using Bowtie2 [86]. Transcript counts were obtained using Samtools [87].

Differential expression analyses were carried out using DEgenes Hunter [88]. This R pipeline examined the relative change in expression between the different samples using different algorithms (EdgeR, DESeq2, Limma and NOISeq) which apply their own normalizations and statistical tests to validate the whole experiment. On the other hand, an analysis of the time course of CaPOD gene expression was performed considering as a reference the expression levels found in green fruits (G). Raw data are accessible at the Sequence Read Archive (SRA) repository under the accession number PRJNA668052. This reference pepper fruit transcriptome and differentially expressed (DE) genes among the analyzed ripening stages and the NO treatment involved the analysis of 24 biological replicates corresponding to 5 replicates of each stage, except for green fruits, which involved 4 replicates.

### 4.6. Fruit Extracts, Protein Assay, Protein Enrichment Using Ammonium–Sulfate [(NH_4_)_2_SO_4_] Fractionation and In-Gel Isozyme Profile of NADP-ME Activity

To obtain the crude extracts, fresh and intact green pepper fruits were used. The pericarps of the fruits were manually cut into longitudinal strips and then into small pieces of approximately 0.5 cm^3^. The pericarp sample was homogenized with a mortar and pestle in the presence of the extraction buffer at a 1:1 (*w*/*v*) ratio. Extraction buffer contained 50 mM Tris-HCl, 0.1 mM EDTA, 1 mM MgCl_2_, 0.1% (*v*/*v*) Triton X-100 and 10% (*v*/*v*) glycerol. After homogenization, the extracts were filtered through two layers of nylon and centrifuged at 30,000× *g* for 20 min at 4 °C. Protein concentration was determined using the Bio-Rad protein assay (Hercules, CA), with bovine serum albumin as standard. 

Protein enrichment using ammonium–sulfate was carried out as previously described in [42]. Briefly, solid (NH_4_)_2_SO_4_ was slowly added to the 30,000× *g* supernatant of green pepper crude extracts up to a saturation of 50% (*w*/*w)*. The solution was kept at 4 °C for about 30 min and then re-centrifuged at 30,000× *g* for 30 min, and the supernatant obtained was saturated to 75% (*w*/*w*) via the slow addition of solid (NH_4_)_2_SO_4_. The mix was centrifuged again, and the protein-enriched pellet obtained (50–75% fraction) was re-suspended in 0.1 M Tris-HCl buffer, pH 8.0, 1 mM EDTA, 0.1% (*v*/*v*) Triton X-100, 10% (*v*/*v*) glycerol. For 12 mL of the original pepper extract, the corresponding pellet was suspended in 1 mL. The enriched 50–75% protein fraction was loaded on a PD-10 desalting column containing Sephadex™ G-25, which enables high-molecular-weight substances (Mr > 5000) to be separated.

Non-denaturing PAGE was carried out in 6% acrylamide gels (19:1, acrylamide/bis-acrylamide ratio) using a Mini-Protean Tetra Cell (Bio-Rad, Hercules, CA, USA). Pepper fruit crude extracts were added to 0.006% (*w*/*v*) bromophenol blue dye and then loaded onto gels. The NADP-ME isozymes were visualized by incubating the gels in a solution consisting of 50 mM Tris-HCl, pH 7.6, 0.8 mM NADP, 5 mM EDTA, 2 mM MgCl_2_, 0.24 mM nitro blue tetrazolium, and 65 mM phenazine methosulfate containing 10 mM malate. When blue formazan bands appeared, the reaction was stopped by immersing the gels in 7% (*v*/*v*) acetic acid.

## 5. Conclusions

In a previous study on sweet pepper fruits, we studied the presence of NADP-ME activity as well as its regulation during ripening and in the presence of certain modulating molecules [41,42]. However, to our knowledge, there is no information on the number of genes that code for the pepper NADP-ME isozymes, which of them are expressed in fruits and how they are modulated during ripening. Accordingly, the present data respond to these questions. We found that pepper fruits express four *CaNADP-ME* genes that are differentially regulated during ripening and in the presence of exogenous NO. Likewise, four NADP-ME isozymes were identified in green fruits (NADP-ME I to NADP-ME4), where NADP-ME III and IV are the most prominent. However, future studies are necessary to unravel the role of each NADP-ME isozyme in fruit ripening, considering that both malate and the reducing power generated during their reaction in the form of NADPH are involved in numerous metabolic processes that undergo continuous adaptive changes during fruit ripening.

## Figures and Tables

**Figure 1 plants-12-02353-f001:**
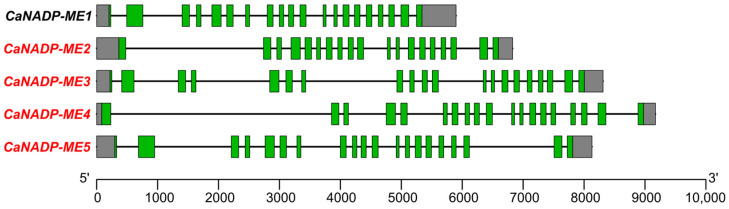
Genomic organization of the pepper *CaNADP-ME* gene family. The structure of the genes is shown with exons indicated by green boxes, and introns are shown as black lines. Untranslated regions are shown by gray boxes. Exon–intron regions are drawn at scale.

**Figure 2 plants-12-02353-f002:**
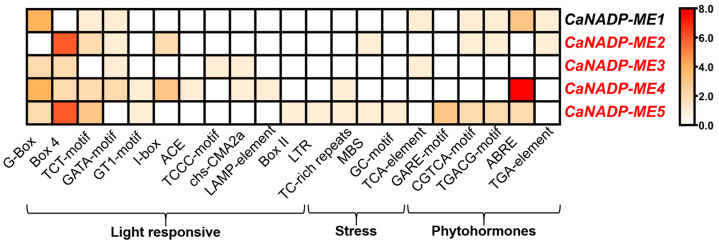
Heatmap of cis-regulatory elements corresponding to the 1500 bp region upstream of the transcription starting point of *CaNADP*-*ME* genes. The cis-regulatory elements were grouped according to their functional implications, including DNA, regulation and cell cycle, light response, stress and phytohormones. Motifs were identified in the PlantCARE database.

**Figure 3 plants-12-02353-f003:**
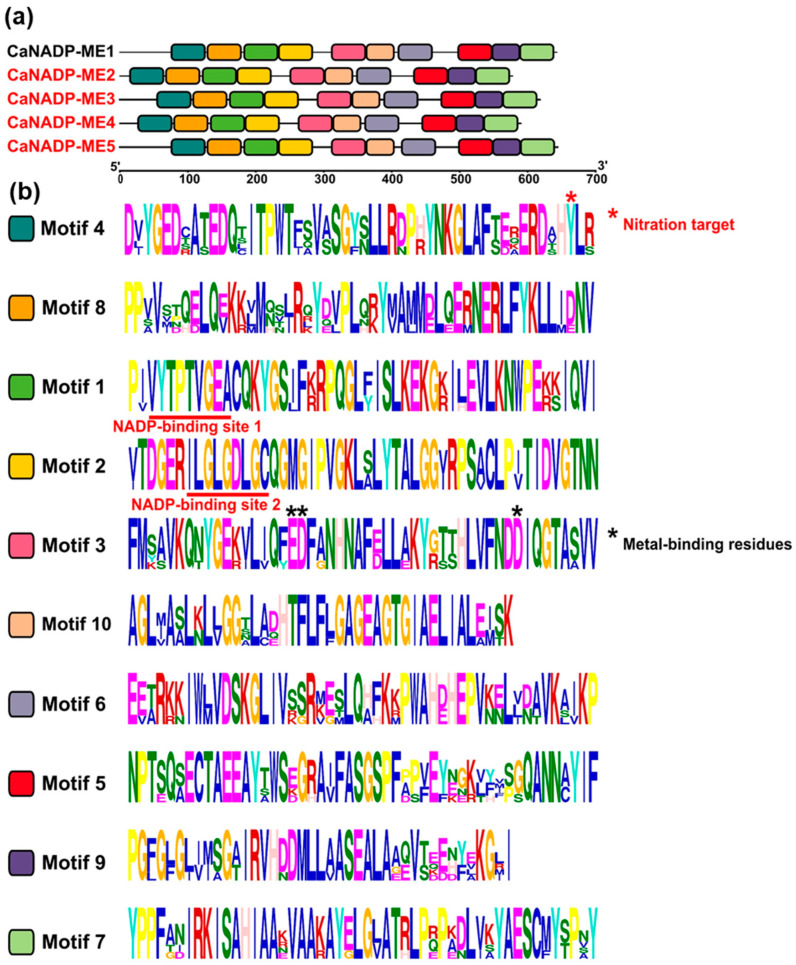
Identification and position of consensus amino acid motifs for pepper CaNADP-MEs. (**a**) Distribution of conserved motifs. The distribution of conserved motifs, numbers 1–10, are represented by boxes of different colors. (**b**) Amino acid sequences of the motifs. Ten amino acid motifs of various sizes were identified. The height of each amino acid symbol is proportional to the degree of conservation in the consensus sequences depicted in the ten motifs. Sequence logos of conserved motifs were created with MEME. The red asterisk in motif 4 denotes the amino acid which is potentially target of nitration, and the black asterisk in motif 2 labels the metal-binding residues.

**Figure 4 plants-12-02353-f004:**
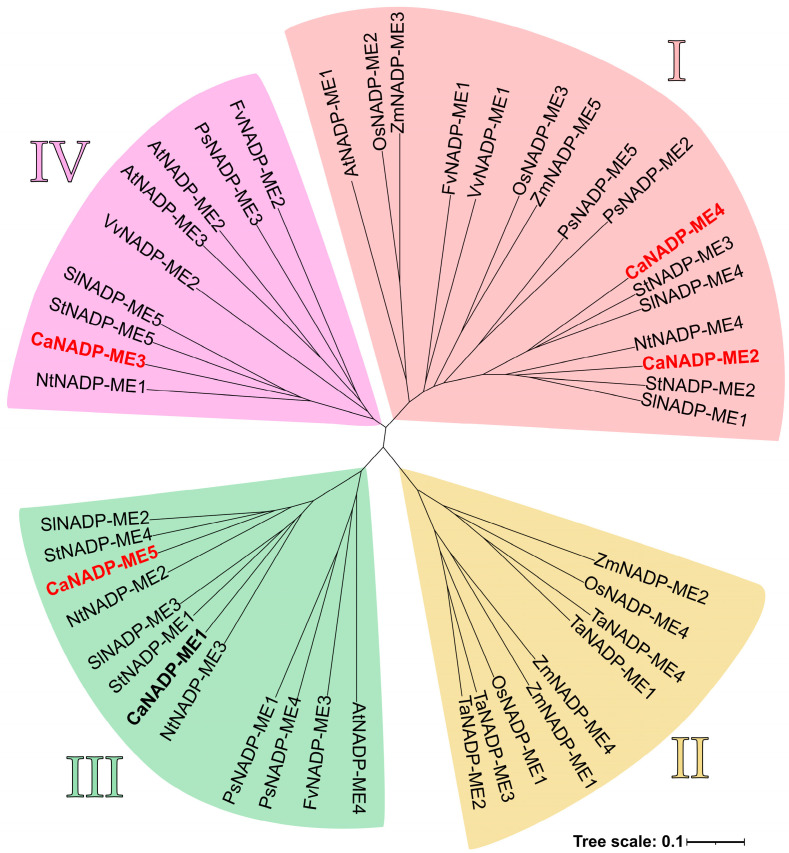
Phylogenetic relationships among NADP-MEs from different plant species. Clusters (I–IV) are displayed using different colors. The NADP-MEs identified in pepper are highlighted in bold. Those found specifically in sweet pepper fruit are indicated in red. Species abbreviations: At (*Arabidopsis thaliana*), Ca (*Capsicum annuum*), Fv (*Fragaria vesca* subs. *vesca*), Nt (*Nicotiana tabacum*), Os (*Oryza sativa* subs. *japonica*), Ps (*Pisum sativum*), Sl (*Solanum lycopersicum*), St (*Solanum tuberosum*), Ta (*Triticum aestivum*), Vv (*Vitis vinifera*) and Zm (*Zea mays*).

**Figure 5 plants-12-02353-f005:**
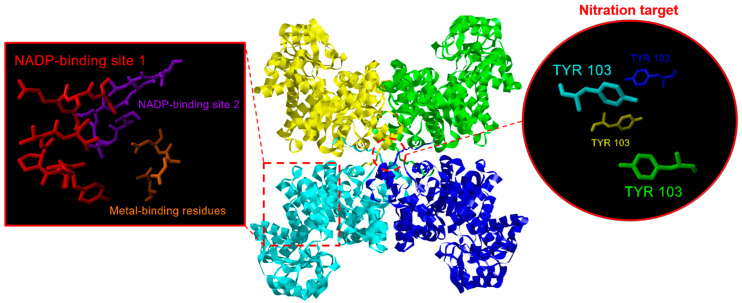
Model of the quaternary structure of CaNADP-ME3 present in pepper fruits. Subunits are colored in yellow, green, cyan and dark blue. On the left, the amino acid residues involved in the binding of NADP and metals in one of the subunits are shown in detail. On the right, the binding site of the subunits is shown, highlighting the Tyr107 residues, which are considered nitration targets.

**Figure 6 plants-12-02353-f006:**
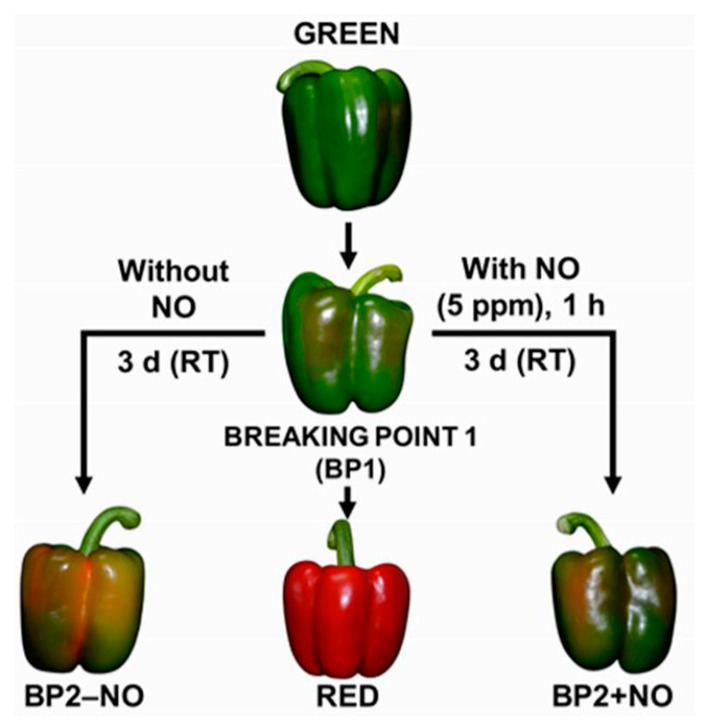
Illustrative picture showing the experimental design used in this study with the representative phenotypes of sweet pepper (*Capsicum annuum* L.) fruits at different stages and treatments: immature green, breaking point 1 (BP1), breaking point 2 without NO treatment (BP2 − NO), breaking point 2 with NO treatment (BP2 + NO) and ripe red. Fruits were subjected to an NO-enriched atmosphere (5 ppm) in a hermetic box for 1 h and were then stored at room temperature (RT) for 3 days. Reproduced with permission from González-Gordo et al. [35].

**Figure 7 plants-12-02353-f007:**
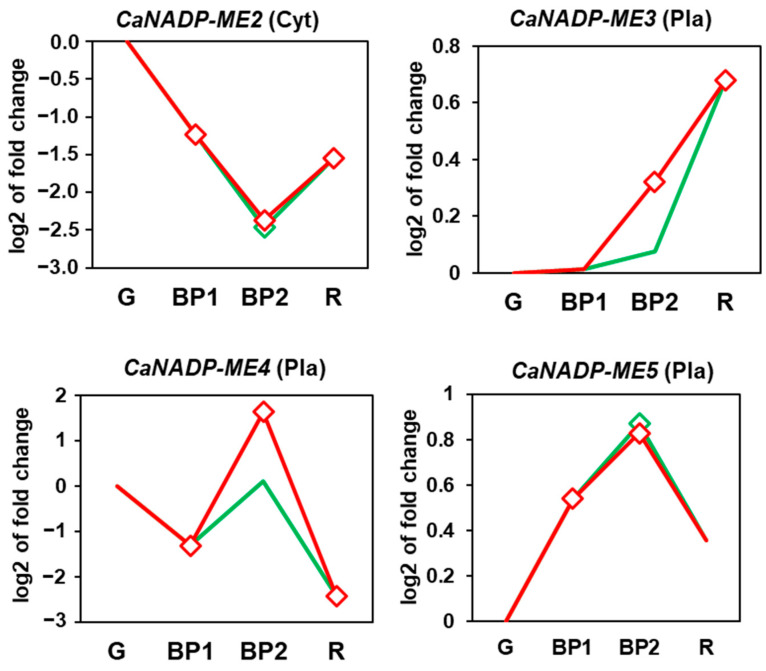
Time course expression analysis of four *CaNADP-ME* genes (RNA-Seq) under natural ripening conditions and after exogenous NO treatment. Samples of sweet pepper fruits at different ripening stages correspond to immature green (G), breaking point 1 (BP1), breaking point 2 with (green line) and without (red line) NO treatment (BP2 + NO and BP−NO, respectively) and ripe red (R). Statistically significant changes in expression levels (*p* < 0.05) compared to green fruit (G) are indicated with diamonds.

**Figure 8 plants-12-02353-f008:**
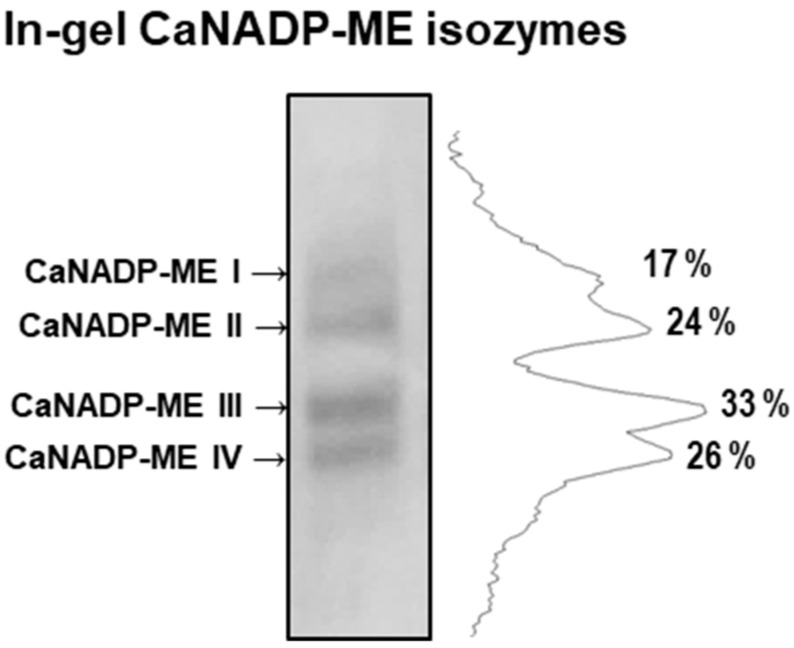
Isozymatic NADP-ME activity in green sweet pepper fruits in an enriched 50–75% ammonium–sulfate protein fraction. Protein samples (240 µg per lane) were separated by non-denaturing polyacrylamide gel electrophoresis (PAGE; 8% acrylamide), and activity was detected using the NBT method (**left**). NADP-ME isozymes were labeled I–IV. Densitometric analysis of these isozymes and their relative quantification (%) were conducted using the ImageJ program according to the isozyme profiles obtained (**right**).

**Table 1 plants-12-02353-t001:** Summary of the five *NADP-dependent malic enzyme* (*NADP-ME*) genes identified in the pepper (*C. annuum* L.) genome and some of the properties related to the protein encoded for these genes, including the number of amino acids (aa), molecular mass (kDa), theoretical pI and their putative subcellular localization. The four *CaNADP-ME* genes specifically detected in the sweet pepper fruit transcriptome are highlighted in red.

Gene Name	Gene ID	Chr.	Genomic Location	Protein ID	Length (aa)	Da	Theoretical pI	Subcellular Localization
*CaNADP-ME1*	107861795	3	267357814–267362949	XP_047266087.1	643	70,840	5.94	Plastid
* CaNADP-ME2 *	107843507	5	158139423–158145651	XP_016543302.1	578	64,245	5.80	Cytosol
* CaNADP-ME3 *	107847755	8	26364857–26372640	XP_016547705.1	618	68,468	6.57	Plastid
* CaNADP-ME4 *	107842298	9	199489875–199498766	XP_016541547.1	590	65,360	6.13	Plastid
* CaNADP-ME5 *	107850438	12	97276100–97283614	XP_016550447.1	644	70,709	6.28	Plastid

## Data Availability

Sequence Read Archive (SRA) data are available at the following link: https://www.ncbi.nlm.nih.gov/sra/PRJNA668052 (accessed on 28 May 2020).

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
