# Peer review of "NADP-Dependent Malic Enzyme Genes in Sweet Pepper Fruits: Involvement in Ripening and Modulation by Nitric Oxide (NO)"

_plants, 2023, doi:10.3390/plants12122353_

Round 1

Reviewer 1 Report

Authors have mentioned that they have identified 5 genes of NADP-ME from genome of pepper, however they have used the proteome data. The tissue/stage of proteome data is also not mentioned. like authors have identified only 4 isozymes in fruits, it is very likely that they may miss some genes from proteome. please explain this properly.

CaNADP-ME3 and CaNADP-ME5 were upregulated, CaNADP-ME2 and CaNADP-ME4 were downregulated in expression? what is their probable role in fruit ripening?

for protein purification and isozyme analysis, authors have used only one stage of fruit, They should have performed studied on different stages to get better picture of protein expression and fruit ripening.

adequate

Reviewer 2 Report

The article "NADP-dependent Malic Enzyme genes in sweet pepper fruits: Involvement in ripening and modulation by nitric oxide (NO)" is well written and the experiment was well executed. The article provides new insight into and information on the CaNADP-ME system which helps to explore new pathways in ripening during exogenous NO treatments. Although few changes are required:

The abstract needs to be more concise, and it explains a lot, authors should add the take-home message at the end of the abstract.

The introduction is well written but the objective of the study has some deficiencies. 

The results and discussion are well written and explain the results in detail.

I would recommend adding supplementary Fig 2 in the main article (not as a supp file).

Overall the article has the potential to be published in the journal.

Minor changes are required.
